# A Novel Approach to Obtain PAR with a Multi-Channel Spectral Microsensor, Suitable for Sensor Node Integration

**DOI:** 10.3390/s21103390

**Published:** 2021-05-13

**Authors:** Eiko Bäumker, Daniel Zimmermann, Stefan Schierle, Peter Woias

**Affiliations:** Laboratory for the Design of Microsystems, Department of Microsystems Engineering—IMTEK, University of Freiburg, 79110 Freiburg im Breisgau, Germany; daniel.zimmermann@neptun.uni-freiburg.de (D.Z.); stefan.schierle@imtek.uni-freiburg.de (S.S.); woias@imtek.de (P.W.)

**Keywords:** photosynthetically active radiation, PAR, light quality, light sensor, ams AS7341, spectrometer, multi-channel spectrometer, photosynthetic photon flux density

## Abstract

We propose a novel approach to measure photosynthetically active radiation (PAR ) in the form of photosynthetic photon flux density with an inexpensive, small multi-channel spectrometer sensor, with integrated optical filters and analog-to-digital converter. Our experiments prove that the combination of eight spectral channels with different optical sensitivities, such as the sensorchip in use (AS7341, ams), derive the PAR with an accuracy of 14 μ$m^{-2}$$s^{-1}$. Enabled by the sensor architecture, additional information about the light quality can be retrieved which is expressed in the reduced light quality index. A calibration method is proposed, and exemplary measurements are performed. Moreover, the integration in a solar-powered wireless sensor node is outlined, which enables long-term field experiments with high sensor densities and may be used to obtain important indexes, such as the gross primary production.

## 1. Introduction

Central Europe has endured two climatically extreme years, including severe drought. This has lead to drastic reductions in crop yields and dieback events in forests [1]. Such incidents disturb not only the local ecosystem and its microclimate formed by the crop canopy but also globally with regard to its carbon exchange with the atmosphere. Monitoring, analyzing, and developing of predictive models for forest evolution are essential to understand the underlying processes, develop strategies to maintain plant health, and monitor the photosynthesis rate. Several indexes to quantify the forest performance exist, such as gross primary production (GPP), which is used to quantify the photosynthesis of organic compounds from carbon dioxide. A widely adopted model of GPP is determined as a function of incident photosynthetically active radiation (PAR), the PAR fraction absorbed by vegetation $f_{PAR}$, and the photosynthetic efficiency *e*. Together, they describe the relation between absorbed energy and bound carbon [2].

For the GPP calculation, it is preferable to obtain all three values (incoming PAR, $f_{PAR}$, and *e*) to achieve a worldwide data acquisition solution. In fact, this is already possible with satellite based approaches to detect photosynthesis at the leaf level, but leaves uncertainties at the ground level [3]. Another approach to quantify the parameters in the forest is to record PAR with quantum sensors. In Reference [2], five quantum sensors are installed stationary above and below the canopy to determine both the incident and absorbed PAR needed for the GPP. Stationary setups utilize towers [4] to enable long-term monitoring, but they only cover a small portion of a forest due to their fixed position and expensive setup. Handheld devices, such as the LI-190 (LI-COR, Inc., Lincoln, Nebraska USA), can circumvent this financial problem and increase the number of measurement spots. However, researchers or skilled personnel are needed to periodically move the sensors and perform measurements. This is impractical and expensive for long-term studies and does not cover large areas. From our point of view, the drawbacks of the system with handheld devices could be circumvented with sensors that are small, cheap, self-sustaining and easy to deploy. Sensors with such properties can be installed at numerous places (in the range of hundreds) within the observed area. Such systems, when equipped with wireless communication, are often referred to as “smart dust” and are running on batteries or solar cells over long periods of time. It may cover areas significantly larger compared to the manual approach, although not as large as a satellite based systems. On top, it would enable the user to study light intensity at different canopy levels to derive the GPP when deployed in horizontal and vertical space.

Beside the light intensity being an important factor in plant growth, the light quality plays an important role, as well, but is not reflected in PAR. The light quality considers the spectral composition of the incident light. It is usually described in a ratio of the integrated intensities of blue (400 nm to 500 nm), green (500 nm to 600 nm), and red (600 nm to 700 nm) intervals [5,6]. Evidence of the influence of the spectral composition is commonly collected in laboratory experiments. For example, plants were exposed to various light quality scenarios but at the same intensity, showing a distinguishable growth behavior [7,8,9]. Plants may heavily differ in the thickness of their abaxial-face epidermis and leaf thickness when exposed to different light spectra [7]. Shengxing et al. have studied the effect of light in the red or blue regimes on biologically inactive and active photoreceptors, such as phytochrome phototropin [10]. Other studies examine the plant growth under varying illumination conditions for different species [11,12]. These studies were conducted in either greenhouse or laboratory facilities, but similar effects have already been observed in studies outside a controlled laboratory environment, such as the study by Solomakhin and Blanke, in which apple orchards were covered by differently colored hailnets, and a change in plant and fruit growth was observed [13].

The light quality is usually measured with expensive spectrometers that can also be used to determine the PAR [6,14]. However, the nanometer precise data set gathered by these devices is not always needed to obtain information about the light quality and is, therefore, often reduced to the three light regimes of blue, green, and red. In such cases, we see the opportunity for this to also be achieved with a inexpensive sensor chip in contrast to the usual approach of obtaining the light quality with a high resolution spectrometer.

To measure PAR and determine the light quality, we have designed, built and tested an inexpensive sensor, using a commercially available spectrometer sensor. While this publication focuses on the sensor itself, including the calculation and calibration methods required for its application, we will also give a brief overview about its capability of being embedded into a wireless sensor node system. Equipped with solar cells providing the needed energy for data acquisition and wireless transmission, it can operate energy-autonomously. Furthermore, the overall size is comparable to a matchbox. This means, the system can be easily attached to small branches simplifying the deployment process and enabling the user to deploy the sensor nodes in a wide range of vertical and horizontal space, making it possible to obtain the GPP. These sensor nodes can be equipped with additional sensors to collect valuable parameters affecting plant growth beside the light intensity and quality. While our sensor node sticks to temperature and humidity data, other sensor combinations to quantify, for example, CO_2_-concentrations or stem diameters are possible and could be integrated in separate sensor nodes and be connected via wireless communication to the overall-system, as well.

## 2. General Sensor Specifications

This study focuses on the design of an inexpensive sensor for photosynthetically active radiation with regard to a later integration in a solar-powered wireless sensor node. The sensor system should provide accurate information about the environmental conditions that are relevant for plant growth. The use case is long-term studies with a node density of more than three nodes per 100 m^2^. Consequently, costs and easy deployment, as well as power consumption and weather resistance, are important aspects of the design. Besides the light intensity in the form of PAR, information about the incident light spectrum should be provided. With regard to the photosynthetic activity of plants, temperature and humidity are commonly recorded by wireless sensor nodes and are also integrated in the proposed system. However, they are not further discussed in this publication. The size and weight of the sensor-node plays an important role in the deployment at different canopy levels, e.g., by attaching it to smaller branches, as required. We have found the size of a matchbox (50 mm × 35 mm × 15 mm) for the full system and a weight below 50 g to be convenient. To the knowledge of the authors, there are no commercially available PAR sensors that could be integrated in such a small sensor device. Size-wise, the PAR sensor MQS-B from the company *Walz* with its cylindrical shape, a 14 mm diameter, a height of only 16 mm, and a weight of 32 g would fit into such a system, but it lacks a weatherproof housing and the ability to be integrated in a sensor node. Furthermore, it does not offer information about the light spectrum, and with a price exceeding 100 euro, costs rapidly increase when numerous nodes should be deployed.

PAR sensors that potentially fit into the desired sensor node have been built in a do-it-yourself approach by Caya et al. [15], Rajendran et al. [16], and Fielder et al. [17] who provide detailed instructions. Their approach uses a photodiode, together with a diffuser and optical filters, to match the sensor response to the desired PAR spectrum. This solution lacks the potential to gather data about the light quality and requires a careful choice of the optical filters. Both quantities can be obtained with a spectrometer, as done by Botero et al. [14]. However, this approach can not be miniaturized to fit into the desired dimensions and would also exceed a reasonable budget per sensor node.

Our approach is to use a commercially available sensor that provides at least three channels distinct in their spectral sensitivity while, as a whole, covering the full 400 nm to 700 nm PAR spectrum. With additional information about the spectral distribution in the red, green, and blue region, the light quality can be further evaluated. The output of sensors with a higher spectral resolution than the three regions could be down-converted to the desired three intervals. Based on this concept, a number of standard RGB or CCD sensors and spectral color sensors can be considered as alternatives for sensing PAR and light quality. As these sensors are commonly used in mobile devices for imaging, spectral identification, and color matching applications, their price and size are reasonably low and, due to the integrated optical filters, costs and effort for external periphery are cut down significantly. In addition, the sensor candidates should fulfill the requirements for an integration into the planned sensor node: First, the sensor needs a high dynamic range, permitting a measurement of very low and high light intensities. It has been reported that the PAR, expressed in photosynthetic photon flux density (PPFD), can get as low as 3 μmol $m^{-2}$$s^{-1}$ below the canopy [18] and over 1600 μmol $m^{-2}$$s^{-1}$ with clear sky and direct sunlight [19]. Popular devices, such as the LI-250A from LI-COR, can cover this range with up to 1999 μmol $m^{-2}$$s^{-1}$ at a resolution of 0.1 μmol $m^{-2}$$s^{-1}$ [20]. A resolution of 1 μmol $m^{-2}$$s^{-1}$ seems reasonable from our point of view for a low cost alternative, while covering the full possible PPFD range. Second, the achievable accuracy for the PPFD measurement should be high. The mentioned LI-250A promises an accuracy of about 0.4% of the reading, neglecting the accuracy of the PAR sensor itself. Barnes et al. have shown that the performance of quantum-based PAR sensors heavily varies at non-uniform light spectra and tends to underestimate the PPFD [21]. A difference of up to 7% was observed even for calibrated devices. Based on this, we aim for a maximal measurement error of 10% in our sensor design.

### 2.1. The ams AS7341 as a PAR and Light Quality Sensor

Although sensors with only three channels in the PAR spectrum would already provide insights on the light quality, quite a number of potential candidates, such as the RGB-sensor APDS-9151 (Broadcom Inc., San Jose, CA, USA) [22], are not sensitive in the desired wavelength regions. Therefore, we decided to use an 11-channel spectrometer sensor, the AS7341 (ams AG, Premstaetten, Austria). This sensor is normally used in mobile devices for spectral identification and color matching applications. Eight of the eleven channels are sensitive in the visible spectral range between 390 nm to 750 nm [23], which matches closely to the PAR spectrum, as shown in Figure 1. It is worth mentioning that due to the inbuilt optical filters and analog-to-digital converter (ADC), additional analog circuitry is not needed which makes the integration very easy and convenient at a sensor cost below 10 Euros. For each channel, the AS7341 returns a count indicating the amount of light that was collected during one measurement. The maximum count number is dependent on the user-modifiable integration time for one measurement interval and, therefore, determines the achievable resolution. A third parameter, the gain of the ADC, controls the sensitivity of each channel. The data can be read out digitally via a common I²C protocol. The sensor footprint of 3 mm by 2 mm and a low current consumption of only 210 μA at 1.8 V (during a measurement) are beneficial for being operated in a compact housing and for a long-term operation powered by solar cells or batteries. In addition, the sensor can be powered down between measurements, resulting in a standby current draw of only 0.7 μA at 1.8 V. For the experiments, the sensor has been embedded in a 3D printed housing shown in Figure 2, which is inspired by typical PAR quantum sensors, such as the LI-190. An acrylic light diffuser plate is chosen to uniformly illuminate the used sensor chip. The housing top with the diffuser is shaped in a way to perform a cosine correction for the incident light. A glass dome is fitted on top of the diffuser to avoid any dirt contaminating the light path.

### 2.2. Embedding the Sensor into an Energy-Autonomous Sensor Node

The new measurement approach with the low cost ams AS7341 as a PAR and light quality-sensor makes it possible to integrate all required hardware and software into a small mobile sensor node, together with other useful sensors, such as for temperature and humidity. This system could be deployed at different canopy levels. In fact, our team has already built a prototype of a wireless sensor node that fulfills the following properties: Firstly, it is attachable to smaller branches, e.g., with a zip tie as the prototype weights only 39 g with dimensions below a typical matchbox as shown in Figure 3. Second, it is self-sustaining. The sensor-node is powered with an array of high-performance solar cell and should be able to run for periods longer than several years without the need to change any primary batteries. The low energy consumption even allows the prototype to work in very poor light conditions as low as 5.5 Wh$m^{-2}$ per day. Third, it provides an easy deployment. The prototype uses the wireless communication technology LoRa-WAN that offers a free-field range of several kilometers at a low power consumption. When there is no public network available, own gateways can be deployed in the forest to receive the data from hundreds of nodes. A remote configuration is also possible due to the bidirectional communication protocol.

## 3. Obtaining PAR from a Color-Matching Spectrometer

There are several possibilities to quantify PAR. The most common one is the photosynthetic photon flux (PPF), also described as the PPF density, called PPFD. It considers all incident photons per time and area, independent of their wavelength, in the range from 400 nm to 700 nm [21]. An alternative definition is the yield photon flux (YPF) which takes the plant growth behavior more closely into account. Here, each photon in the range of 360 nm to 760 nm is weighted according to its wavelength and the relative quantum efficiency curve of plants. Various measurement principles exist to obtain one or both of PPFD and YPF. A first option is the usage of a spectrometer, such as the popular LI-1800. These usually measure the spectral power distribution (SPD) over the full PAR range with a spectral resolution often better than 1 nm. The PPFD and YPF are then calculated by integrating the acquired SPD throughout the PAR spectrum and are, in case of the YPF, multiplied with a defined weighting. This principle allows an accurate PAR result even for narrow-band light sources. The second option is a quantum sensor, using a photo-diode and an optical band-pass filter for the PAR wavelength region. The photocurrent is given by the integral over the product of diode responsivity and incoming optical power at a given wavelength. Due to the non-ideal photodiode responsivity with respect to the desired sensitive area in the PAR region, these sensors are less accurate for narrow-band light sources compared to the approach using a spectrometer [21]. As the photocurrent does not allow to deduce the sensed spectrum, only the PPFD can be monitored. With the lack of any spectral information and the non-ideal diode response curve, careful calibration is needed for the desired operational scenario, i.e., the correlation of the PAR to the GPP.

For our proposed sensor, we are using an integrated circuit (IC) with a 4×4 photodiode array and two separated ones with dedicated functions builtin. The photodiodes in the array are paired, and their output is internally combined, integrated, and digitized. To determine the PPFD, it is enough to sum all integrated photocurrents generated by the channels that are sensitive in the PAR range. The overall spectral response can be optimized by weighting each diode output. Along with the technical effort, this method provides a reasonably accurate spectral resolution, of about 30 nm to 50 nm, which enables the user to detect scenarios with unfavorable light conditions that negatively influence the correlation between PPFD and GPP. It is conceivable that even a coarse YPF could be determined evaluating the channel signals individually. At least, the detection of an unevenly distributed light spectrum provides auxiliary information.

The rough spectral information that is provided by the sensor can be used as a simplified measure for light quality. Since the sensor is not able to provide a nanometer precise spectral resolution, we consider the obtainable data set as reduced light quality index (RLQI). The RLQI can be a measure of light condition while obtaining the PPFD in parallel. It divides the PAR spectrum into three equally sized intervals for red (700 nm to 600 nm), green (600 nm to 500 nm), and blue (500 nm to 400 nm). The RLQI is expressed as the fraction of all diode outputs in the corresponding interval over the total sum. It is convenient to express the spectral distribution over the three regions as a percentage of the overall intensity, such as 20%-35%-45%, in the order of red-green-blue (r-g-b).

## 4. Experimental Setup

The following experiments aim to demonstrate and investigate the feasibility of using a sensor chip with 8 channels in the PAR region to obtain the PPFD and the RLQI. For the PPFD measurement, the PAR data of our sensor system is compared with a reference (LI-190). The RLQI was determined when the AS7341 was illuminated by a light source with a known emission spectrum. The channel counts of the corresponding spectral interval were compared to the SPD of different light sources. All experiments were performed with the AS7341 embedded in its developed housing. The ADC was configured with an integration time of 100 ms with a gain of four for all eight channels in the visible spectrum and a gain of one for the remaining IR channels. The settings are chosen in a way that the channel outputs are about half of the maximum possible count number on a cloud-less bright summer day. The settings are not changed throughout all experiments to circumvent the necessity of a re-calibration.

For the calibration of our sensor to determine the PAR, we have used the method of referencing the measured data to a calibrated reference sensor, similar to the open-sky tests from Fielder et al. [17], since an optical radiation calibrator was not available. The AS7341 was exposed to different light scenarios together with a reference PAR sensor (LI-190 with LI-250A light meter). Both the eleven output channels of the AS7341 and the reference PPFD were recorded under solar radiation outside and in an insulator box with artificial light. Outside, 31 data-sets at different weather and canopy situations were measured and taken into account for the PAR calibration procedure. The measured values are in the range of 1.7 μmolm^−2^s^−1^ to 594 μmolm^−2^s^−1^. In the insulator box, a set of LEDs with narrow-band emission and two standard halogen 40 W and 75 W light bulbs were used as light source. The set of LEDs consisted of 12 LEDs of the manufacturer LUMILEDS, each emitting at a non-overlapping spectral region with a half-width of about 30 nm. The specification of the used “L1C1”-LED series can be found in their datasheet [24]. Two white LEDs with 2700 K and 5700 K were also used beside the light bulbs as two broadband light sources. All LEDs were facing both sensors with a distance of 100 mm. The selected LEDs were used to perform a coarse spectral sweep over the PAR region in the steps according to the LED center wavelengths by using them individually. The obtained 19 data sets in the insulator box cover a PAR range from 0 μmolm^−2^s^−1^ to 45 μmolm^−2^s^−1^ with a mean value of 14 μmolm^−2^s^−1^.

For the determination of the RLQI, the data set was extended by two additional measurements of a warm and cold white LED. With the known SPD of each LED, it was verified qualitatively that the ratio of the AS7341 channel output corresponds to the expected distribution.

## 5. Results

An exemplary test in the insulator box where the AS7341 was exposed to one of three narrow-bandwidth LEDs with a peak emission in the violet, green, and red spectrum confirms that the output of the individual channels, shown in Figure 4, is according to their advertised sensitivity. The usage as a PAR sensor by combining them should, therefore, be feasible.

### 5.1. Determination of the PAR

Our approach is to use a linear model to calculate the PPFD from the raw output of the AS7341 channels. With the obtained data, including the PPFD of the reference, we use multiple linear regression (MLR) to find the parameters of a linear conversion function of the form
(1)$$y_{i}=b_{0}+\sum_{i=1}^{n}\left(b_{i}x_{i}\right)+\epsilon .$$

In Equation (Equation 1), each channel output of the AS7341 corresponds to a single explanatory variable $x_{i}$. The reference value of the PAR sensor acts as the dependent variable ($y_{i}$). The slope coefficients $b_{i}$ for each explanatory are then determined with the MLR. The set of 50 measurement points and usage of the eight channels lead to the derived slope coefficients summarized in Table 1. The model has an adjusted coefficient of determination
of 0.991 and a root-mean-square error of 16 μmolm^−1^s^−2^. Both values indicate that the usage of the AS7341 output is suitable to calculate the PAR with a simple MLR algorithm.

However, the experimental data set is not ideal to achieve a representative model built via the MLR. Low light conditions are over-represented as shown in Figure 5. When measuring outside under clear sky conditions, a PPFD in the range of 300 μmolm^−1^s^−2^ to 550 μmolm^−1^s^−2^ was determined. This matches the results of Chiang et al., who have been measuring the PPFD during a diurnal course in Basel, Switzerland, with a spectrometer in November 2018 [6]. For a calibration that is valid over the whole possible light intensity range, further calibration is needed at higher PPFD. The determined residuals are slightly lower, except for some outliers in data sets gathered outside where only sunlight is recorded (Figure 5) compared to those taken in the box. This may be due to the usage of very narrow-band LEDs in the insulator box, which we expect were not measured accurately by our PAR reference.

In addition to the MLR model, using the eight channels sensitive in the PAR wavelength region, all other possible combinations, including the NIR and clear channels, are used to calculate a regression model for the PPFD. The models are then sorted by their adj. *R*^2^ and RMSE and a f-test is performed. This test is indicating whether the model is valid beyond our samples. All combinations showed a low p-value below 0.01, stating that there is a good linear relationship between the channel output and the PPFD. The lowest adj. *R*^2^ of 0.994 and RMSE of 12 μmolm^−1^s^−2^ is obtained for the solely inclusion of the channels one, three, six, eight and further the NIR. This is remarkable, as the near infrared channel is, as said, only sensitive in the infrared. In an ideal case, the NIR output should be negligible, since, according to the datasheet, the reference PAR sensor is not sensitive in this region. The correlation between NIR and PAR indicates that there is still responsivity of the reference PAR sensor in that region. That would explain why large errors with PPFD sensors are found in Reference [21] for narrow-band near infrared light sources.

### 5.2. Reduced Light Quality Index

The data set used to determine the PPFD is applied here. The channels (as can be seen in Figure 4) can be combined to the desired interval determining the ratios of counts in the red, green, and blue light region, thus providing a measure for determining the RLQI. To examine the approach of adding all color interval channels, a cold-white and a warm-white LED with known emission spectrum were investigated. The experimental data is plotted in Figure 6. The output of each channel is represented as a bar centered at the expected center wavelength of the channel and a width according to the channel’s FWHM. The values are taken from the data sheet [23]. The counts are normalized to the highest count number in the data set. The expected spectral emission of the light source is shown by the gray dashed line. It has been reconstructed by a cubic interpolation of data points extracted from the graph in the LED data sheet. The given PPFDs are measured with the reference sensor and our regression model. The channel counts provide a relatively good representation of the expected spectra. For the cold-white LED (LUXEON C 5700K/90, LUMILEDS), the characteristic peak in the blue wavelength region is visible, as well as the broadband maximum in the red-green area. Similar observations can be made for the warm-white LED (LUXEON C 2700K/80, LUMILEDS). The counts in the blue region are reduced and the broadband peak is red-shifted. The warm-white LED’s RLQI was determined to be 43%-47%-10% (red-green-blue), with a maximum absolute deviation of 6% from the expected distribution. For the cold-white LED, an RLQI of 35%-45%-20% (r-g-b) was measured. Here, the deviation is in the range of 13% points. This is caused by the sensor channels sensitive in the blue interval, which seem to under-represent the narrow peak in the blue of the 5700 K LED spectrum. In addition, no spectral analysis of the used light sources was performed; therefore, we rely only on the accuracy of the data sheet.

Nonetheless, in our opinion this is enough evidence that the principle for determining the RLQI can be applied in the field. To provide a first demonstration, the PPFD was measured in the case of an open blue sky (15 September 2020, in Rheinfelden, Germany, see Figure 7a) and under some leaves (Figure 7b) with comparable ambient conditions (30 July 2020, in Freiburg, Germany). The RLQI changes visibly from 29.8%-40.6%-30.6% (r-g-b) under blue sky to a spectrum where the blue part is reduced with respect to the green and red region, resulting in 34.4%-45.4%-20.2% (r-g-b). This shows that there is a definite loss in information of the spectral distribution if only the PAR is considered for plant growth and photosynthesis estimation in different illumination scenarios. The light quality may also vary during the day, with changing weather conditions and may even be different at various canopy levels in the vegetation or plantation. For the clear sky scenario, our measured light quality can also be compared to a the results of Chiang et al. [6], who performed a light quality measurement with a spectrometer in Basel (less than 75 km from our measurement site). The blue fraction ranges from 27% to 40%, green from 31% to 35%, and red from 27% to 39%. These values were measured for different solar angles that were heavily influencing the relation of red and blue fractions of the spectrum. Comparing the values in Figure 7a) to this, it is obvious that the green fraction is higher than expected. The measurement of blue and red parts are within the ranges measured by Chiang et al. Therefore, the proposed RLQI can provide some deeper insight in the spectral distribution of the incident light, while measuring the PPFD.

## 6. Discussion and Conclusions

Our concept suggests to use a cheap and inexpensive multi-channel spectrometer chip (AS7341) to measure photosynthetically active radiation in the form of PPFD. The additional RLQI information about the spectral distribution in the red, green, and blue light regime facilitates a better statement about the validity of a correlation between the measured PAR and the expected GPP. Our experiments prove that the PAR can be derived using the output from eight channels with a mean error of about 14 μmolm^−2^s^−1^.

We have used a multiple linear regression to determine a transfer function of the spectrometer output signals to the PPFD showing a high adj. *R*^2^ of 0.99. This proves that there is a linear correlation between the channel output and the desired PPFD value. Hence, a multi-channel spectrometer can be used for this purpose. The coefficients were obtained through a set of measurements at different lighting conditions. In this data set, low light conditions are over-represented which may negatively influence validity of the model at high PPFD. To increase the accuracy of the sensor, more measurements under real conditions have to be performed.

In addition to the calculated PPFD, the RLQI provides information about the light distribution in the three spectral regions. The digital outputs of the channels provide a good representation for several light scenarios with known spectra. We see a maximum absolute deviation between the expected and measured RLQI of 4%.

Measurements taken below some dense leaves show, as expected, an RLQI with a peak in the green region. Records at clear sky scenarios also agree well with those measured by Chiang et al. Despite this result, care has to be taken in interpreting the RLQI. An exact binning of the spectrum as done by the RLQI can not be accurately achieved with broad-band sensors by design. While the combination of the AS7341 channels fit well to this case, there is, for example, still a slight overlap of the channel sensitivity in the blue and red region, but their output is fully counted to the green region. This results in a slight overestimation of the green incidence. A full compensation of that error is to our knowledge not possible, although reduced when a thorough calibration is done for a known and limited set of light conditions.

The IC ams AS7341 shows that a cheap, commercially available spectrometer can be used as PAR and light quality sensor. It is able to cover the desired PAR wavelength region between 400 nm and 700 nm and required intensities from 0 μmolm^−2^s^−1^ to 1600 μmolm^−2^s^−1^. The introduced RLQI accounts for the fact that theses sensors do only provide a limited spectral resolution, which, however, is sufficient for a lot of applications. In fact, its ability to supply additional spectral information beyond the PAR region, as done with the near infrared channel, can provide valuable information with regard to the plant growth. It is worth noting that the additional channel in the infrared region improves the accuracy of PAR determination via the MLR according to our experiment. As the channel’s sensitivity lies outside the PAR region, there should not be any contribution. It has to be clarified whether this is due to imperfections of the used reference sensor LI-190 or the AS7341 itself, being still sensitive in the not desired NIR region. A further improvement in accuracy can be done by examining the different sensitivity and integration settings, the AS7341 provides. In our experiments only one fixed setting that covers the full range of light intensity was used but a more flexible approach could possibly further improve sensitivity for PPFD and RLQI at conditions with very low or high light intensity. The AS7341 is an ideal candidate that can be deployed in applications with a high number of sensors. The integrated optical filters and ADC provide a simple digital interface that facilitates the development of a complete PAR and RLQI sensor due to minimal needed additional periphery. Its cost, size, weight, and power consumption make it perfect to be used in a wireless sensor node that can be massively deployed in the field.

We have already built a solar powered wireless sensor node that uses the AS7341 including additional sensors for temperature and humidity. For a field and long-term study, a larger and improved number of sensor devices are currently prepared and will be tested in a real field scenario to gain experience and data how well such a device can be used. We are confident that the demonstrated device can improve the data quality recorded in the field. With numerous sensor nodes deployed at different canopy levels and large areas, not only a higher granularity is possible, but the system itself can either become an alternative or supplement method for manual or satellite-based recording. The data can expedite a deeper understanding of biological and environmental processes in a forest helping to reduce the effort needed to measure common indexes, such as the GPP.

## Figures and Tables

**Figure 1 sensors-21-03390-f001:**
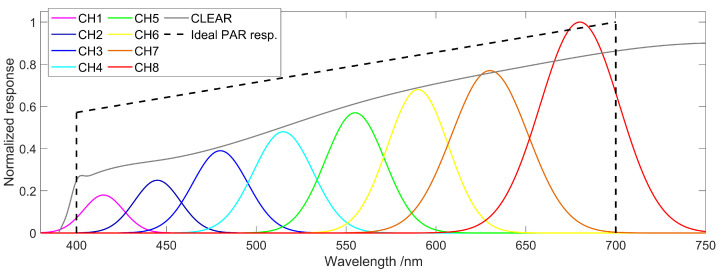
Spectral responsivity according to the AS7341 datasheet [23] (graph redrawn with permission). The eight channels show the whole PAR spectrum. Not shown is a channel with a sensitivity in the near infrared at 910 nm.

**Figure 2 sensors-21-03390-f002:**
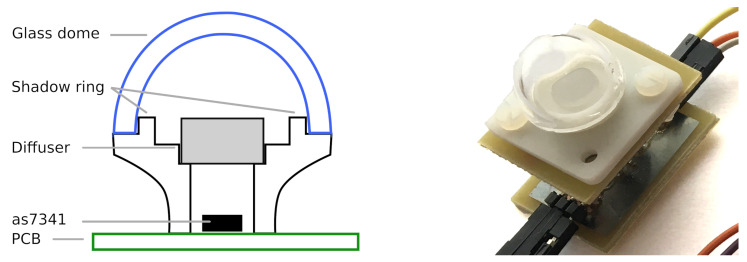
For all tests, the AS7341 is put into a housing formed to achieve cosine correction for incoming light. As the optical filters are integrated in the AS7341, there is no need for additional parts in the housing other than a diffuser and in our case an optional glass dome.

**Figure 3 sensors-21-03390-f003:**
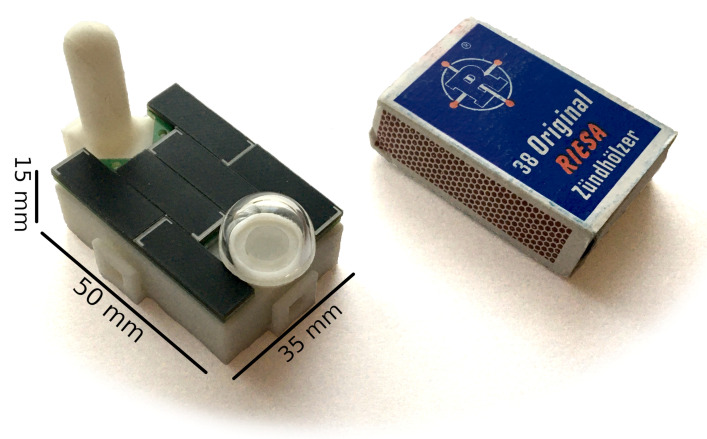
Sensor node prototype with embedded AS7341 and circuitry, in comparison to a standard matchbox as a dimension reference. On top are six solar cells, the glass dome over sensor and the antenna.

**Figure 4 sensors-21-03390-f004:**
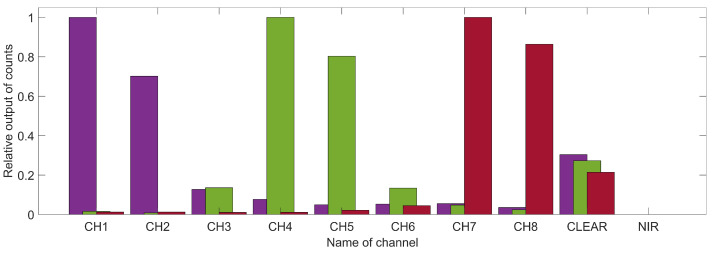
Three example data sets measured with the AS7341 in the insulator box. In each set, either a narrow-bandwidth LED with violet ( 425 nm), green ( 530 nm), or red ( 629 nm) emission (in the figure shown from left to right) is turned on. As expected, channels corresponding to the LED spectrum show a high signal. The AS7341’s clear channel shows an output for every LED due to its broadband sensitivity. The near infrared (NIR) channel, sensitive in the near infrared, shows as expected negligible output.

**Figure 5 sensors-21-03390-f005:**
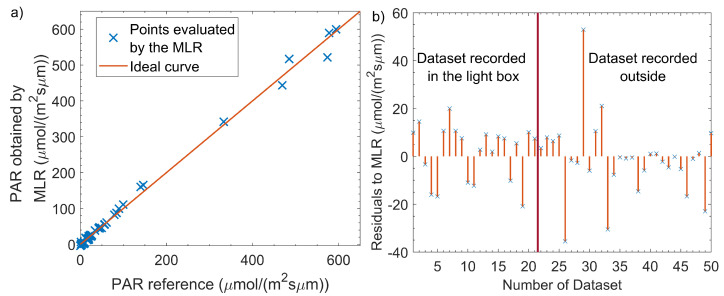
(**a**) Distribution of the obtained PAR via the LI-190 reference and predicted values of the MLR using data from the AS7341. The recorded data sets are over-represented at low light conditions. (**b**) Residuals of the MLR for each data set.

**Figure 6 sensors-21-03390-f006:**
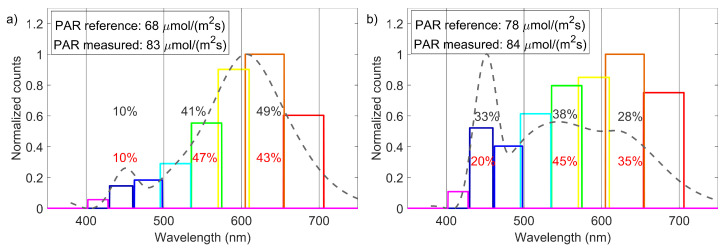
Measured spectral light distribution of a warm (T=2700 K) white (**a**) and a cold (T=5700 K) white (**b**) LED, normalized to their highest channel value. The bar widths correspond to the FWHM of the channel sensitivity. The gray-dashed line represents the expected spectrum of the light source (redrawn from the data sheet [24]). The measured and calculated PAR values are listed in the upper left box. The RLQI distribution that is expected is noted in gray, the measured one in red.

**Figure 7 sensors-21-03390-f007:**
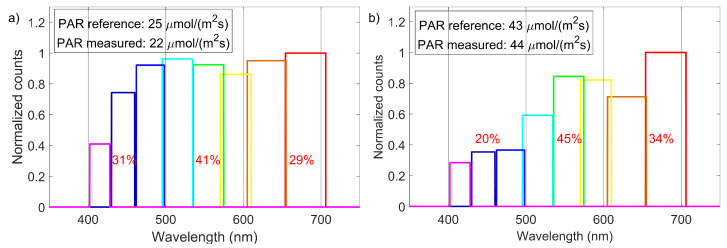
Measurement of direct sunlight on a clear summer day (**a**) and under a leaf (**b**). The measured PAR s with both sensors are noted in the graph. The spectral channels are normalized to the highest count number of all channels. The RLQI values do not sum up to 100% because of rounding errors.

**Table 1 sensors-21-03390-t001:** Resulting slope coefficients of the multiple linear regression including the AS7341’s channels F1–F8 for the complete data set.

*b* _0_	*b* _1_	*b* _2_	*b* _3_	*b* _4_
−9.8853	0.0046	0.0136	0.0243	0.0459
***b*** **_5_**	***b*** **_6_**	***b*** **_7_**	***b*** **_8_**	
−0.0471	0.0195	0.0178	−0.0026

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
