# Peer review of "A Novel Approach to Obtain PAR with a Multi-Channel Spectral Microsensor, Suitable for Sensor Node Integration"

_sensors, 2021, doi:10.3390/s21103390_

Round 1

Reviewer 1 Report

General comment

In the article “A novel approach to obtain PAR with a multi-channel spectral microsensor, suitable for sensor node integration”, the authors propose a small and cheap multichannel spectrometer sensor to measure PAR. I have no doubt that the development of a miniature sensor for PAR measurement, such as the device presented in the paper, is commercially attractive and could have a very good field of application in daily agronomic practices. However, I do not find the work innovative enough to be published in a journal such as Remote Sensing.

Comments on concepts

 In addition to the technical description of the new device, the author introduce a new concept: the “reduced light quality index”. In my opinion, this issue is not clearly presented and explained.

Units

micromol photons m-2 s-1 instead of micromol m-2 s-1

Figures

In general the font size used for figure legends is too small (they should be readable at 100% zoom).

Figure 1. Change colour from blue to black for the ideal PAR response or change the line style.

Figure 5. Delete excess spaces when using parentheses. In general, there should be a space before an open parenthesis and no space after it, and there should be a space after the closed parenthesis and no space before it.

Figures 6b and 7. Avoid overlapping between text and bars.

Written language

There are many gramar and spelling errors in the text. It should be proofread by an English native speaker.

Some examples below:

L. 5, 46, 172, 178 PAR and not the PAR

L. 18 “Leaf area index” and no “leave area index”

L. 48 …“ and they nevertheless have to be validated with a long-term ground truth” ?

L. 59 “relatively expensive” instead of “relative costly”

L.76 “weather resistance” and not “a weather resistance”

L. 178 by PAR measurements

Author Response

Dear Reviewer,

We would like to thank you for the thoughtful comments and efforts towards improving our manuscript.

Point 1: I have no doubt that the development of a miniature sensor for PAR measurement, such as the device presented in the paper, is commercially attractive and could have a very good field of application in daily agronomic practices. However, I do not find the work innovative enough to be published in a journal such as Remote Sensing.

Response 1: You are mentioning the choice of the of the journal - we have been thinking about this question as well, since we know, that this manuscript is of a rather technical nature. However, we found the idea and prove that a commercially available sensor can be easily adopted to fit the user’s needs when collecting PAR data of larger areas highly interesting for the readers of Remote Sensing.
With the today's low boundaries for developing own small electronic sensor systems (the popular open-source hardware platform Arduino is a great example here) chances are that the audience may want to adopt and further advance our development.

We are planning to publish information about the underlying hardware and source code after the sensors have been thoroughly tested, to enable alternative sensor solutions in the fields of ecology, environmental sciences, and civil engineering.

Point 2: In addition to the technical description of the new device, the author introduce a new concept: the “reduced light quality index”. In my opinion, this issue is not clearly presented and explained.

Response 2: Thanks to your comment, we clarified the concept of the reduced light quality index with our corrections. The light quality index is usually determined by evaluating the light conditions with a spectrometer. We wanted to emphasize by the ‘reduced’, that the sensor is not capable of measuring at nanometer precise spectral resolution and can not replace a standard spectrometer when a more precise measurement is needed.

Point 3: Figures & written language

Response 3: Concerning your grammar and graph formatting advises:
We are especially thankful for your advice on the graph formatting. We have changed them according to your suggestions.

We hope that your comment concerning typos and grammar mistakes could be addressed in a satisfying manner.

Summary of our changes:
- Rewrote introduction: Now giving more detailed explanations about the application and a focus on GPP instead of the LAI

- RLQI: Included references to a group reporting comparable results

- Conclusion: Slight changes in conclusion for PAR. Added a comparison of the calculated PAR with expected results reported by another group.

- Figures: Made font larger, minor changes in legends

- Language: Thorough proof reading was done, and errors were corrected

Reviewer 2 Report

This paper shows a significance impact to others research, cheap and durable sensor is a future.  

Some typo in line 213 was found "()"

Author Response

Dear Reviewer,

We would like to thank you for the thoughtful comments and efforts towards improving our manuscript. We changed some parts of the manuscript, especially the introduction, to improve the quality of the paper. The typo you found is corrected in the final version.

Summary of our changes:
- Rewrote introduction: Now giving more detailed explanations about the application and a focus on GPP instead of the LAI

- RLQI: Included references to a group reporting comparable results

- Conclusion: Slight changes in conclusion for PAR. Added a comparison of the calculated PAR with expected results reported by another group.

- Figures: Made font larger, minor changes in legends

- Language: Thorough proof reading was done, and errors were corrected

Reviewer 3 Report

Dear Editor in Chief of Remote Sensing Journal,

In the paper entitled “A novel approach to obtain PAR with a multi-channel spectral microsensor, suitable for sensor node integration”, the authors propose an inexpensive sensor to measure photosynthetically active radiation (PAR) and discuss calculation and calibration methods of the designed sensor. Moreover, they provide an overview on the sensor’s capability to be embedded into a small and energy-autonomous wireless sensor node. It is concluded that the newly introduced inexpensive sensor is well-capable of measuring PAR with reasonable accuracy compared to the traditional expensive sensors.

The authors have done a good job in sensor design and building an inexpensive sensor for PAR estimation. The manuscript is generally well written and properly organized except the introduction part which needs to be significantly rewritten with exact justification and explanation on PAR and Leaf Area Index (LAI) estimation from ground and satellite based measurements. The introduction part, as it is now, includes some inexact information about LAI retrieval. Moreover, the justification for building a new inexpensive sensor and its applications is not properly explained, where the focus is put on PAR (and LAI) estimation under a forest canopy for remote areas. The results part and comparisons can also be further expanded to different conditions and comparisons with available sensors and methods.

My real concern, however, is about the technical details/aspects of the manuscript that makes me doubtful if Remote Sensing is the right place to publish this paper and the general interest of the journal readers towards such technical papers. I assume that other journals, for instance Sensors might be a better choice to publish this paper.

To conclude, I would recommend the manuscript for publication with moderate revisions if the technical aspects of the work suits the scope of the journal.

More specific comments follow below.

L28: please change “Leave area index(LAI)” to “Leaf Area Index (LAI)”

From a reader’s standpoint, I think the introduction is somehow trying to convey the message that LAI can only be estimated from the 0.4-0.7 micron (chlorophyll sensitive part) and using light intensity measurements, although this is true for what we call ‘Green LAI’, total LAI can actually be estimated from any other part of the spectrum as long as there are absorption lines for vegetation canopy such as VNIR and SWIR since LAI is the total one-sided area of all leaves (green and brown). I would suggest to rewrite the introduction with clearly stating this issue to avoid any misunderstanding.

L51: “This is especially needed when it is desired to calculate the LAI from the radiative transfer in the canopy”, I didn’t understand what the authors want to say by this! Please clarify. RT modelling and LAI retrieval from RT is totally different and has nothing to do with ground measurements, unless for calibration purposes.

In general, I suggest to rewrite the introduction part in a better structure and avoid giving inexact information. Although, the current experiments can be considered enough for the first publication, the results are very limited for a newly introduced sensor and need to be expanded and includes many real world scenarios in future works.

Author Response

Dear Reviewer,

We would like to thank you for the thoughtful comments and efforts towards improving our manuscript. Your detailed review of our manuscript and the nice feedback regarding the sensor design is very helpful and is further encouraging us to pursue improving the sensor and performing long-term studies with it.

Point 1: The introduction part, as it is now, includes some inexact information about LAI retrieval. Moreover, the justification for building a new inexpensive sensor and its applications is not properly explained, where the focus is put on PAR (and LAI) estimation under a forest canopy for remote areas. The results part and comparisons can also be further expanded to different conditions and comparisons with available sensors and methods.

Response 1: We are especially thankful for your advice to rewrite the introduction. We did so and we hope that the structure is now easier to follow and the points we wanted to make are described more precisely. The indeed inexact information about the LAI is hereby removed.

Point 2: My real concern, however, is about the technical details/aspects of the manuscript that makes me doubtful if Remote Sensing is the right place to publish this paper and the general interest of the journal readers towards such technical papers. I assume that other journals, for instance Sensors might be a better choice to publish this paper.

Response 2: To address your concerns about the choice of the journal: We have been thinking about this question as well, since we know, that this manuscript is of a rather technical nature. However, we found that the idea and prove that a commercially available sensor can be easily adopted to fit the user’s needs when collecting PAR data of larger areas highly are of interest for the readers of Remote Sensing.
With the today's low boundaries for developing own small electronic sensor systems (the popular open-source hardware platform Arduino is a great example here) chances are that the audience may want to attend and further advance our development.
We are planning to publish information about the underlying hardware and source code after the sensors have been thoroughly tested, to enable alternative sensor solutions in the fields of ecology, environmental sciences, and civil engineering.

Summary of our changes:
- Rewrote introduction: Now giving more detailed explanations about the application and a focus on GPP instead of the LAI

- RLQI: Included references to a group reporting comparable results

- Conclusion: Slight changes in conclusion for PAR. Added a comparison of the calculated PAR with expected results reported by another group.

- Figures: Made font larger, minor changes in legends

- Language: Thorough proof reading was done, and errors were corrected

Round 2

Reviewer 1 Report

The authors have properly addressed the reviewers´comments and the new version is actually improved.

I have some very minor comments: 

L. 39-40 "deploy" is repeated. If possible find a synonym.

L.43-44 "cover" is repeated. Try to look for a synonym.

L.70-72 Please revise the whole sentence

"To measure PAR and determine the light quality, we have designed, built and tested an inexpensive sensor that is capable of measuring both PAR and a light quality, using a commercially available spectrometer sensor"

the highlighted part should be deleted. It is reiterative.

L. 80 "can be" is repeated

L. 280 margin

L.379 margin

Author Response

Dear Reviewer,

Thank you very much for your comments and the thorough proofreading of the manuscript. We appreciate your commitment to improve our manuscript.
We revised our paper with regard to your minor comments:

Point 1: L. 39-40 "deploy" is repeated. If possible find a synonym.
Replaced by “installed”

Point 2: L.43-44 "cover" is repeated. Try to look for a synonym.
Rephrased whole sentence

Point 3: L.70-72 Please revise the whole sentence
Reiterative part deleted

Point 4: L. 80 "can be" is repeated
Is not corrected

Point 5: Margins:
This error was introduced by the highlighting software and will not happen in the non-highlighted version.